# Identification of Novel Cyanopyridones and Pyrido[2,3-*d*]pyrimidines as Anticancer Agents with Dual VEGFR-2/HER-2 Inhibitory Action: Synthesis, Biological Evaluation and Molecular Docking Studies

**DOI:** 10.3390/ph15101262

**Published:** 2022-10-13

**Authors:** Tarfah Al-Warhi, Al-Aliaa M. Sallam, Loah R. Hemeda, Mahmoud A. El Hassab, Nada Aljaeed, Ohoud J. Alotaibi, Ahmed S. Doghish, Mina Noshy, Wagdy M. Eldehna, Mona H. Ibrahim

**Affiliations:** 1Department of Chemistry, College of Science, Princess Nourah bint Abdulrahman University, Riyadh 84428, Saudi Arabia; tarfah-w@hotmail.com (T.A.-W.); noaljaeed@pnu.edu.sa (N.A.); ojalotaibi@pnu.edu.sa (O.J.A.); 2Department of Biochemistry, Faculty of Pharmacy, Badr University in Cairo (BUC), Badr City 11829, Egypt; alia.sallam@pharma.asu.edu.eg (A.-A.M.S.); ahmed_doghish@azhar.edu.eg (A.S.D.); 3Biochemistry Department, Faculty of Pharmacy, Ain-Shams University, Abassia, Cairo 11566, Egypt; 4Department of Medicinal Chemistry, Faculty of Pharmacy, Beni-Suef University, Beni-Suef 62514, Egypt; loah_pharma@yahoo.com; 5Department of Medicinal Chemistry, Faculty of Pharmacy, King Salman International University (KSIU), SouthSinai, Ras Sudr 46612, Egypt; 6Biochemistry and Molecular Biology Department, Faculty of Pharmacy (Boys), Al-Azhar University, Nasr City 11231, Egypt; 7Department of Clinical Pharmacy, Faculty of Pharmacy, King Salman International University (KSIU), SouthSinai, Ras Sudr 46612, Egypt; mina.noshyaziz@yahoo.com; 8Department of Pharmaceutical Chemistry, Faculty of Pharmacy, Kafrelsheikh University, Kafrelsheikh 33516, Egypt; 9School of Biotechnology, Badr University in Cairo, Badr City 11829, Egypt; 10Department of Pharmaceutical Medicinal Chemistry and Drug Design, Faculty of Pharmacy (Girls), Al-Azhar University, Cairo 11884, Egypt; mona.hussein@azhar.edu.eg

**Keywords:** cyanopyridone, pyridopyrimidine, antitumor activity, dual VEGFR-2/HER-2 inhibitor, molecular modeling

## Abstract

In the current work, we designed and synthesized three families of non-fused and fused compounds based on cyanopyridone: derivatives of 6-amino-1,2-dihydropyridine-3,5-dicarbonitrile (**5a-f**) and 3,4,7,8-tetrahydro pyrimidine-6-carbonitrile (**6a-b** and **7a-e**). The newly synthesized compounds’ structure were determined using a variety of techniques, including ^1^H NMR, ^13^C NMR, mass spectrum, infrared spectroscopy, and elemental analysis. The developed compounds were tested for the ability to inhibit the growth of breast adenocarcinoma (MCF-7) and hepatic adenocarcinoma (HepG2) cell lines using MTT assay. Some of the synthesized compounds were more effective towards the cancer cell lines than the standard treatment taxol. The best antiproliferative activities were demonstrated by non-fused cyanopyridones **5a** and **5e** against the MCF-7 cell line (IC_50_ = 1.77 and 1.39 μM, respectively) and by compounds **6b** and **5a** against the HepG2 cell line (IC_50_ = 2.68 and 2.71 μM, respectively). We further explored **5a** and **5e**, the two most potent compounds against the MCF-7 cell line, for their ability to inhibit VEGFR-2 and HER-2. Finally, docking and molecular dynamics simulations were performed as part of the molecular modeling investigation to elucidate the molecular binding modes of the tested compounds, allowing for a more thorough comprehension of the activity of compounds **5a** and **5e**.

## 1. Introduction

Cancer is a disease that not only represents significant threat to people’s health all over the world but also represents a significant obstacle for those working in the field of medicine [1,2]. It is essential, prior to the development of new potential anticancer candidates, to identify molecular targets that participate in both the growth and death of cancerous cells. Epidermal growth factor receptor (EGFR/ERBB, class I RTK) and vascular endothelial growth factor receptor (VEGFR, class V RTK) are subfamilies that control fundamental cell behavior. HER2-TK is one of the most significant tyrosine kinases and, despite its inability to interact directly with its ligand, it is a favored dimerization partner that can create strong heterodimers with other EGFR/ERBB protein family members [3,4,5,6]. At the cell surface, the normal expression of HER2-TK is important for cell growth regulation and epithelial cell survival. On the other hand, aggressive metastatic breast cancer and other types of cancer frequently show overexpression and amplification of the HER-2 oncogene [7,8]. Similarly, VEGFR-2 plays an important role in the regulation of angiogenesis and survival [9]. Several inhibitors of HER-2 or VEGFR-2 have achieved significant clinical benefits in cancer management. Interestingly, better antitumor activity was observed when combining both HER-2 and VEGFR-2 inhibitors [10,11].

In several clinically used and under-investigation small molecules, including the anti-infective fluoroquinolone drugs and the carbamoyl pyridone HIV-1 integrase inhibitors, the fused and non-fused pyridine motif has been presented as a core structure [12,13]. Drugs based on 2-pyridone, in particular, have been approved for use in a number of other therapeutic fields, including oncology, cardiovascular, respiratory, and CNS disorders [14,15,16,17]. Moreover, kinase inhibitors, such as trametinib [18], palbociclib [19], duvelisib [20], and ripretinib [21], are frequently found to contain the 2-pyridone motif. Sorafenib **I**, which contains a pyridine moiety, is a potent multi-kinase inhibitor that drastically reduced the VEGFR2 transcription in HepG2, HeLa, and A549 cells [22]. In addition, neratinib **II**, an irreversible pan-HER kinase inhibitor, contains a cyanopyridine component that has shown therapeutic effectiveness in HER-2-positive and HER-2-mutated breast tumors [23]. 3-Cyanopyridine derivative **III** showed high cell-growth-inhibitory effects against human MCF-7breast cancer, NCI-H460non-small cell lung cancer, and SF-268CNS cancer cell lines (IC_50_ = 0.02, 0.01, and 0.02 µg/mL, respectively) [24]. Furthermore, the 2-amino-3-cyanopyridine derivative **IV** showed anti-proliferative activities against HCT116 and Huh7 cell lines (Figure 1). It also adopted a reasonable pose to bind to the ATP-binding site of VEGFR-2 kinase in its inactive DFG-intermediate conformation [25]. Meanwhile, pyrimidines and fused pyrimidines are important components of nucleic acids. Further, many pyrimidine-based derivatives show significant antitumor and antiproliferative activities [26,27]. Previously, Pfizer reported that compound **V** with a benzo-fused pyrimidine ring possessed anti-HER-2 activity lower than 42 nM [27]. Further, fused pyridopyrimidine analogs are well-known heterocyclic compounds used in cancer chemotherapy [28,29]. For example, the pyridopyrimidine analog **VI** was identified as a potent VEGFR-2 inhibitor (Figure 1).

Based on the above observations and due to the multifactorial nature of human malignancies, synthesizing a single molecule having different modes of action could be beneficial in treating of different tumors. Accordingly, there is a real need to explore new compounds with simultaneous inhibition of VEGFR-2 and HER-2 receptor tyrosine kinases, with the aim of enhancing antitumor activity, decreasing side effects, and, potentially, overcoming the drug resistance against existing drugs.

## 2. Rationale and Design

In the field of cancer therapy, the synthesis of novel pyridone-based derivatives continues to receive a lot of attention. Several studies have shown that 2-pyridones are an efficient isostere of pyridines and that they exhibit a similar level of potency [30,31,32,33]. However, in comparison to pyridines, 2-pyridones have the advantage of being able to act as both hydrogen bond donor and acceptor. Because of this, 2-pyridone based molecules are featured as kinase inhibitors because they have attractive binding to kinases at their ATP binding clefts [34,35,36]. In addition, they have the potential to affect the lipophilicity, aqueous solubility, as well as the metabolic stability of the target drug [12].

The target compounds were designed taking into consideration the antiproliferative and VEGFR-2/HER-2-inhibitory activity of several reported pyridine- and pyridopyrimidine-based anticancer molecules. Hence, in our research, it was expected that the newly synthesized 2-pyridone based derivatives might exert an anticancer effect in a similar manner to the above-mentioned biologically active pyridines **II** and **IV** with enhanced binding affinity (Figure 2). This study aimed to explore two new sets of 3-cyanopyridone-based derivatives. The 3-cyano group was thought to contribute in the perfect binding to the active sites by creating a vital polar hydrogen bond interaction [37,38].

Given that many active anticancer compounds integrate the trifluoromethyl group as a pivotal motif in their structures [39,40,41], the trifluoromethyl phenyl moiety was incorporated and proposed to be essential for anticancer activity in the newly developed series. Such a group was expected to form a hydrogen bonding interaction with the amino acid residues involved in the active sites of VEGFR-2 and HER-2. Further, the phenyl groups were thought to interact with the hydrophobic region in both targets. Hence, the 4-phenyl-3-cyanopyridone bearing the trifluoromethyl phenyl moiety was kept as the core structure in the two new series. The first series had various 4-phenyl substitutions in the cyanopyridone ring in compounds **5a-f**, with the aim of attaining a better understanding of the SAR and discovering more potent dual target HER-2 and VEGFR-2 inhibitors (Figure 2). Furthermore, to verify the significance of the bicyclic ring system with cyanopyridone and pyrimidine moieties for the antitumor activity and to explore their possible anticancer mechanism, new series **6a-b** and **7a-e,** with or without the 2-methyl substitution on the pyrimidine ring, were synthesized through ring closure at the 5-cyano and 6-amino groups of several 3-cyanopyridone derivatives (Figure 2).

## 3. Results and Discussion

### 3.1. Chemistry

The synthetic routes for the target molecules (**5a-f**, **6a-b**, and **7a-e**) are shown in Figure 1 and Figure 2. The first batch of target molecules, 2-oxo-pyridine-3,5-dicarbonitriles **5a-f**, was produced in a one-pot reaction of cyanoacetanilide 3 [42] with a range of aromatic aldehydes **4a-f** and malononitrile in ethyl alcohol using a catalytic quantity of piperidine [43] (Figure 1).

IR spectra of 2-oxo-pyridine-3,5-dicarbonitriles **5a-f** revealed the characteristic forked absorption bands of amino, cyano, and carbonyl functional groups around 3250, 2210, and 1656 cm^−1^, respectively. The ^1^H NMR spectra for the latter molecules disclosed the presence of a D_2_O-exchangeable singlet signal attributable to the protons of the NH_2_ functionality around *δ* 4.45–8.26 ppm. Further, the ^1^H NMR spectrum of cyanopyridone **5f** displayed two singlet signals at *δ* 3.77 and 3.85 ppm corresponding to the protons of the three OCH_3_ groups.

The target tetrahydropyrido[2,3-*d*]pyrimidine-4,7-dione derivatives **6a-b** were synthesized by cyclocondensation of 2-oxo-pyridine-3,5-dicarbonitriles (**5b** and **5d**) with formic acid/H_2_SO_4_ [44], whereas their 2-methyl analogs **7a-e** were successfully obtained via reaction of 2-oxo-pyridine-3,5-dicarbonitriles **5** with acetic anhydride [45]. The ^1^H-NMR spectra for pyridopyrimidines **6** showed pyrimidine-H_2_ at *δ* 8.33 ppm. The ^1^H-NMR spectra of compounds **7** showed an additional singlet at *δ* 1.93–2.20 ppm due to the newly introduced CH_3_ motif, whereas their ^13^C NMR spectral data disclosed a characteristic signal at *δ* 21.52–22.44 ppm due to the CH_3_ group.

### 3.2. Biological Evaluations

#### 3.2.1. Cytotoxicity Assay

The MTT colorimetric assay is a very sensitive method used to measure the potential cytotoxicity (loss of viable cells) of targeted compounds. It was thus used to determine the IC_50_ for all of the compounds synthesized here (**5a-f**, **6a-b**, and **7a-e**) against the breast cancer (MCF-7) and hepatocellular carcinoma (HepG2) cell lines. The drug taxol was used as the reference standard. Cell morphological changes were recorded to determine the cytotoxic effect, and the results are found in Table 1, Figure 3.

It was noticed that compounds **5a** (IC_50_ values = 1.77 ± 0.1 and 2.71 ± 0.15 µM) and **5e** (IC_50_ values = 1.39 ± 0.08 and 10.70 ± 0.58 µM) showed higher activities than taxol (IC_50_ values = 8.48 ± 0.46 and 14.60 ± 0.79 µM) against both MCF-7 and HepG2 cells, respectively. Compound **7b** showed higher activity (IC_50_ value = 6.22 ± 0.34 µM) than taxol against the MCF-7 cell line. While compounds **5b**, **5c**, **5e**, **6a**, **6b,** and **7c** displayed higher activity (IC_50_ values ranged from 2.68 ± 0.14 to 11.37 ± 0.61 µM) than taxol against the HepG2 cell line, other tested compounds showed poor to moderate activity against the two selected cell lines compared to taxol (Table 1).

The SAR was based on the two selected cell lines and on a comparison of the tested compounds against the reference taxol. It was observed that most synthesized compounds with the non-fused cyanopyridone moiety showed remarkable activity against both cell lines compared to most pyridopyrimidine derivatives.

The unsubstituted phenyl-bearing cyanopyridone derivative **5a** showed the best cytotoxic activity against the HepG2 cell line (IC_50_ = 2.71 ± 0.15 µM) and the second best cytotoxic activity against the MCF-7 cell line (IC_50_ = 1.77 ± 0.10 µM) compared to taxol. The approach to increase the lipophilicity in series **5** by grafting 2,4-dichloro substituent in the phenyl moiety resulted in cyanopyridone derivative **5e**, which exerted the best anticancer activity against the MCF-7 cell line (IC_50_ = 1.39 ± 0.08 µM) in this study. It was found that only the 2,4-dichloro substitution could enhance the anticancer activity within series **5** against MCF-7 cells in comparison to the unsubstituted phenyl-bearing cyanopyridone **5a**, whereas the remaining substitutions (4-OH, 4-NO_2_, 4-Cl and 3,4,5-(OCH_3_)_3_) did not show improvements in the activity against both cell lines. Concerning the impact of substitution of the phenyl moiety within the non-fused cyanopyridones **5a-f**, the activities against MCF-7 cells were decreased in the order of 2,4-(Cl)_2_ > H > 4-Cl > 4-NO_2_ > 4-OH > 3,4,5-(OCH_3_)_3_, whereas the order was H > 4-NO_2_ > 2,4-(Cl)_2_ > 4-OH > 4-Cl > 3,4,5-(OCH_3_)_3_ for the activities against HepG2 cells.

Concerning the fused pyridopyrimidine derivatives, the unsubstituted C-2 on the pyrimidine moiety was deemed to be more favorable than the 2-methyl substitution against HepG2 cells, whereas the 2-substitution on the pyrimidine moiety seemed to be less favorable against MCF-7 cells. In addition, the *p*-phenyl substitution with electron-withdrawing functionalities in **6b** (4-Cl) and **7b-c** (4-Cl, 2,4-di(Cl)_2_) showed greater anticancer activities than the *p*-electron-donating substituted derivatives **6a** (4-OH) and **7e** (4-OCH_3_) against both cell lines. It was noticed that the 4-chlorophenyl pyridopyrimidine **6b** with no 2-methyl substitution on the pyrimidine moiety disclosed the best cytotoxic activity (IC_50_ = 2.68 ± 0.14 µM) among all tested series against HepG2 cells. Noticeably, introduction of 2-methyl substitution on the pyrimidine moiety in compound **7b** resulted in sevenfold lower activity (IC_50_ = 19.58 ± 1.06 µM) than its corresponding derivative **6b** without 2-methyl substitution against HepG2 cells. However, the 4-Cl substitution in **7b** displayed the highest activity (IC_50_ = 6.22 ± 0.34 µM) among all the fused **6** and **7** series pyridopyrimidine compounds and showed the third best activity among all tested compounds against MCF-7 cells.

Clearly, the combination of the 2,4-dichlorophenyl substitution and 2-methylpyrimidine substitution in the derivative **7c** slightly increased its activity (IC_50_ = 9.28 ± 0.50 µM) compared to the unsubstituted derivative **6a** (IC_50_ = 10.68 ± 0.58 µM), while its activity decreased (IC_50_ = 9.28 ± 0.50 µM) relative to the monochloro-substituted derivative **7b** (IC_50_ = 6.22 ± 0.34 µM) against MCF-7. Inversely, the activity of **7c** increased (IC_50_ = 5.04 ± 0.27 µM) in comparison to **7b** against HepG2 cells.

Overall, the impact of substitution of the phenyl moiety in the fused pyridopyrimidines **6a-b** and **7a-e** on the activities against MCF-7 cells showed the order of 4-Cl (2-CH_3_) > 2,4-(Cl)_2_ (2-CH_3_) > H (2-CH_3_) > 4-Cl (2-H) > 3,4,5-(OCH_3_)_3_ (2-CH_3_) > 4-OCOCH_3_ (2-CH_3_) > 4-OH (2-H), whereas the activities against HepG2 cells showed the order of 4-Cl (2-H) > 2,4-(Cl)_2_ (2-CH_3_) > 4-OH (2-H)> H (2-CH_3_) > 4-Cl (2-CH_3_) > 4-OCOCH_3_ (2-CH_3_) > 3,4,5-(OCH_3_)_3_ (2-CH_3_).

#### 3.2.2. VEGFR-2 and HER-2 Kinase Assay

Compounds **5a** and **5e,** which showed the highest antiproliferative activities toward the MCF-7 cancer cell line, were further selected to examine the in vitro inhibitory activities against VEGFR-2 and HER-2 enzymes. The enzyme inhibitory activities (IC_50_) are presented in Table 2 and Figure 4.

As depicted in Table 2 and Figure 4, compound **5e** showed the best inhibitory effect (IC_50_ = 0.124 ± 0.011 µM) on the VEGFR-2 enzyme in comparison to the standard Lapatinib (IC_50_ = 0.182 ± 0.010 µM) and in comparison to compound **5a** (IC_50_ = 0.217 ± 0.020 µM). In a similar pattern, compound **5e** displayed more potent inhibitory effects on HER-2 kinase (IC_50_ = 0.077 ± 0.003 µM) than Lapatinib (IC_50_ = 0.131 ± 0.012 µM) and compound **5a** (IC_50_ = 0.168 ± 0.009 µM).

#### 3.2.3. Impact on the Expression Levels of VEGFR-2 and HER-2 in MCF-7 Cells

In this study, we further explored the impact of cyanopyridones **5a** and **5e** on VEGFR-2 and HER-2 expression levels in MCF-7 cells (Figure 5). The results disclosed that both **5a** and **5e** had interesting profiles for reduction of VEGFR-2/HER-2 expression levels compared to the control cells and the reference drug taxol. Treatment of MCF-7 cancer cells with cyanopyridones **5a** and **5e** significantly down-regulated the expression levels of VEGFR-2 (884.30 and 510.00 pg/mL, respectively) and HER-2 (299.50 and 148.60 pg/mL, respectively) compared to the untreated control (1573.00 and 559.00 pg/mL) (Table 3). Notably, these results are in parallel with the outputs from the cytotoxicity assay, where the dichloro-substituted derivative **5e** was more active than the unsubstituted cyanopyridone derivative **5a**.

### 3.3. Molecular Docking Studies

With respect to the potent antitumor activity demonstrated by the cyanopyridone derivatives synthesized herein, a docking simulation experiment was performed using the most potent cyanopyridones **5a** and **5e**. The docking conducted aimed to furnish valuable insights into the binding modes of compounds **5a** and **5e** with VEGFR-2 and HER-2 enzymes. These insights should, in turn, provide a future guide for further optimizing the proposed compounds toward the ideal anti-breast cancer agents. The molecular operating environment (MOE) 2019.02 was brought into action to conduct and visualize all the docking stages.

Firstly, to validate the docking study, the co-crystalized poses of sorafenib and TAK-285 were re-docked to the active sites of their corresponding targets. The calculated RMSD values between the co-crystalized and the docked poses were found to be 0.52 and 0.77 Å for sorafenib and TAK-285, respectively, which highlights the validity of the applied docking procedures (Figure 6). In addition, sorafenib and TAK-285 achieved docking scores (*S*) of −15.1 and −13.6 kcal/mol, respectively. They established sufficient interactions with the amino acid residues of VEGFR-2 and HER-2, allowing them to dock into the enzyme binding site (ligand affinity). Therefore, achieving a binding mode similar to those observed for sorafenib and TAK-285 was crucial for the enzymatic activity of tested compounds, as was establishing the key interactions with the binding sites, in addition to their acceptable energy scores. Therefore, the previously mentioned scores were used as comparative benchmark values for the docked compounds **5a** and **5e**.

#### 3.3.1. Docking of Compounds **5a** and **5e** into VEGFR-2 Active Site

As Figure 7 reveals, sorafenib binds to the VEGFR-2 enzyme through multiple interactions, and those with critical importance were Cys1045, Asp1046, Glu885, and Cys919. The potent inhibitory activities of compounds **5a** and **5e** are explained by their ability to achieve docking scores of −14.5 and −15.2 kcal/mol, very comparable to sorafenib’s docking score (−15.1 kcal/mol). In addition, both compounds succeeded in achieving the interaction pattern required for inhibiting VEGFR-2. For instance, the triflouro methyl phenyl moiety was engaged in three hydrogen bonds with Cys1045, Asp1046, and Phe1047; moreover, the cyano motif adjacent to the carbonyl of the amide functional was engaged in two hydrogen bonds with Val898 and Ile1044. At last, the amino group formed a hydrogen bond with the key residue Glu885 (Figure 8). Furthermore, both compounds **5a** and **5e** showed similar binding modes and strong interactions with the VEGFR-2 binding site via the ortho chloride atom involved in a hydrogen-bonding interaction with Cys1045. Accordingly, the superior inhibitory activity of compound **5e** over compound **5a** could possibly be attributed to the extra substitution with dichloro atoms.

#### 3.3.2. Molecular Docking Studies of Compounds **5a** and **5e** to Explain Their in Vitro Enzymatic Activity with Her-2 Active Site

The PDB ID 3RCD was used to provide the structural coordinates of HER-2 kinase domain complexed with TAK-285. Figure 9 shows that TAK-285 formed multiple interactions with the residues composing the HER-2 active site. The trifluoromethyl group of the co-crystalized inhibitor formed three interactions with Thr798, Ser783, and Thr862, and the chloride atom interacted with Leu796 through the formation of hydrogen bonds. The nitrogen of the pyrimidine, the carbonyl of the amide, and the hydroxyl functional contributed to hydrogen bond interactions with Met801, Val734, and Gly727, respectively. Interestingly, both compounds **5a** and **5e** achieved more favorable docking scores (−14.1 and −14.5 Kcal/mol, respectively) than the docking score of TAK-285 (−13.6 Kcal/mol) with HER-2. Moreover, the two tested compounds accommodated the necessary interaction pattern within the vicinity of the HER-2 binding site.

As Figure 10 depicts, compound **5a** maintained a strong interaction with the binding site of HER-2 enzyme, forming various interactions. For example, the trifluoromethyl group formed hydrogen bond interactions with Glu770 and Ile767, in addition to two hydrogen bond interactions with Ala771. Moreover, each of the two cyano groups contributed as acceptors for two hydrogen bonds; namely, with Val797, Thr798, Lys753, and Asp863 residues. Finally, the unsubstituted phenyl ring interacted with the key residue Thr862. In alignment with the enzyme assay results, compound **5e** was predicted to have the best docking score and the strongest interaction pattern (Figure 10). The cyano group of compound **5e** adjacent to the amino group formed two hydrogen bonds with Val734 and Lys 753, while the other cyano group formed three hydrogen bonds with Ser783, Leu785, and Thr798. In addition, the carbonyl group engaged in two hydrogen bonds with Asp863 and Thr862, whilst the amino and trifluoromethyl groups interacted through hydrogen bonding with Asp863 and Gly865, respectively. The 2,4-dichlorophenyl moiety played a central role, giving rise to the superior potency and binding affinity of pyridone **5e** through the formation of six hydrogen bond interactions with Val734, Lys 753, Met801, Leu800, Leu852, and Ser783.

In conclusion, the results of the docking analysis showed excellent docking scores and interesting binding modes and interactions of compounds **5a** and **5e** with both VEGFR2 and HER-2 targets, in agreement with their in vitro kinase inhibitory activities. In comparison to sorafenib and TAK-285, compounds **5a** and **5e** were able to maintain strong binding with target receptors, good interaction, and a similar binding mode for VEGFR-2 and HER-2 targets, albeit at slightly different orientations. The best docking results revealed the formation of a highly stable kinase-compound complex for compound **5e**, which showed strong interactions between the two receptors and a wide variety of interesting interactions.

### 3.4. Molecular Dynamics

#### 3.4.1. RMSD and RMSF Analysis

The results retrieved from the biology and docking studies presented compound **5e** as a promising anticancer agent. Therefore, we were encouraged to obtain further in silico insights using molecular dynamic simulations (MDSs) and their applications. Well-established MDSs are privileged over other in silico studies in the precise estimation of the stability of a protein–ligand complex. Accordingly, this advantage was taken into account and the binding poses of **5e** with VEGFR-2 and HER-2, as retrieved from the docking step, were simulated for 150 ns. The Apo form of each target, alongside the binding of each target to its corresponding co-crystalized ligand, was added to the simulation list to provide means of comparison and benchmarking.

As Figure 11A,B show, both free VEGFR-2 and HER-2 proteins demonstrated significant dynamicity, serving their endeavors as primary oncogenic proteins. This was highlighted by RMSD calculations, in which the unbound proteins of VEGFR-2 and Her-2 achieved RMSD values of 5.5 and 4.6 Å, respectively. Compound **5e** demonstrated an excellent ability to restrict the dynamic nature of both VEGFR-2 and HER-2, as evidenced by the lower RMSD values of **5e** with VEGFR-2 and HER-2, nearly 1.9 and 1.8 Å, respectively. The extent of the decrease in RMSD values for **5e** was very close to sorafenib (1.3 Å) and TAK-285 (1.6 Å), respectively. The RMSF values were similar to the RMSD values, in which the residues of the Apo proteins reached average fluctuations of 4.1 and 4.2 Å for HER-2 and VEGFR-2, respectively (Figure 12). The binding of **5e** and sorafenib to VEGFR-2 induced great stability for its residues, lowering their fluctuation to average RMSFs of 1.6 and 1.2 Å, respectively. Similarly, the binding of **5e** and TAK-285 to HER-2 receptor resulted in obvious stability for almost all the residues, as indicated by the significant decreases in RMSF, reaching average values of only 1.7 and 1.6 Å for **5e** and TAK-285, respectively. In general, both the RMSD and RMSF calculations ked to one conclusion: that compound **5e** is capable of potently inhibiting both HER-2 and VEGFR-2 owing to its ability to form strong and stable interactions with the active sites of the two targets.

#### 3.4.2. Binding Free Energy Calculations Using MM-PBSA Approach

This part of the study aimed to provide extra confirmation on compound **5e** as a potential dual HER-2/VEGFR-2 inhibitor. Accordingly, the free-access g_mmpbsa package by Kumari et al. was utilized to define the precise free binding of energy for compound **5e** with HER-2 and VEGFR-2 [46]. To provide a benchmarking means, the free binding of energy between TAK-258 and sorafenib with HER-2 and VEGFR-2 was added to the g_mmpbsa calculations list. All forms of binding free energy were calculated using the last 50 ns trajectories from the production stage. In this context, four types of energies were computed; namely, electrostatic energy, van der Waal energy, polar solvation energy, and SASA energy (Table 4).

As seen in Table 4, the calculated binding free energies for the four complexes were as follows: compound **5e** achieved −290.9± 2.8 and −284.1± 2.6 kJ/mol with HER-2 and VEGFR-2, respectively, while TAK-285 and sorafenib achieved binding free energy of -293.8 ± 2.2 and −313.2 ± 3.5 kJ/mol with HER-2 and VEGFR-2, respectively. These calculations revealed comparable results for the newly synthesized compound **5e** with the crystal references TAK-285 and VEGFR-2. The results also highlighted the importance of the lipophilic nature of the inhibitors, as indicated by the ΔE van der Waal calculations, which were the highest-contributing energy type in the three inhibitors. In this context, it was logical that the two chloride substituents of cyanopyridone **5e** contributed to its superior activity over cyanopyridone **5a**. Moreover, all the MD and energy calculations favored the potentiality of the compound **5e** dual inhibitor for HER-2/VEGFR-2, which was consistent with the earlier enzyme assay results.

The energy contribution of each residue was the final parameter used to investigate the binding effect of cyanopyridone **5e** on the active site residues of HER-2 and VEGFR-2. For this, specific commands were utilized to decompose the total binding free energy into per-residue contributions. As depicted in Figure 13A, the binding of the lead compound **5e** to the active site of HER-2 resulted in favorable contributions to the energy in almost all the residues, especially those interacting with **5e.** For instance, Asp863, Val734 Lys753, Leu785, Thr798, Thr862, and Gly865 were the most favorably contributing residues, indicating the ability of **5e** to form a stable complex within the binding site of HER-2. In a similar way, the binding of cyanopyridone **5e** contributed favorably to the free energy of the surrounding residues in the active site of VEGFR-2, notably with significant contributions for Glu885, Ile1044, Cys1045, Asp1046, and Phe1047 (Figure 13B). In conclusion, all the in silico studies were aligned with each other and the biological investigation, which endorses the ability of cyanopyridone **5e** to act as a dual HER-2/VEGFR-2 inhibitor.

## 4. Experimental

### 4.1. Chemistry

Regular commercial providers were contacted for reagents and solvents, which were then used without further purification. All reactions were routinely examined using Merck Silica Gel 60 F254 (0.25 mm thick) thin-layer chromatography (TLC) and visualisation with a UV lamp. The melting points were determined using the Stuart SMP3 instrument in open capillary tubes. IR spectra were identified using a Shimadzu FT/IR 1650 (Perkin Elmer, Waltham, MA, USA) spectrometer (KBr). Using a Brüker Advance-400 instrument, 1H and 13C NMR spectra in DMSO-d6 were recorded at 400 MHz for 1 H and 101 MHz, respectively. Chemical shifts (*δ*) were measured in parts per million (ppm) in comparison to either the solvent used to record the spectrum or to TMS, which functioned as an internal standard. A Shimadzu GS/MS-QP 2010 Plus spectrometer was used to acquire mass spectra at a wavelength of 70 eV.

#### 4.1.1. General Procedures for the Synthesis of 2-cyano-N-(3-(trifluoromethyl)phenyl) acetamide (3)

After heating a solution of 3-(trifluoromethyl) aniline (1.6 g, 10 mmol) and ethyl cyanoacetate (1.1 g, 10 mmol) at 180–190 °C for 3 h until the release of ethanol stopped, the reaction mixture was cooled, and the precipitate was washed with 100 mL of ether and recrystallized from ethanol to obtain 90% of intermediate **3** [47], melting point = 136–138 °C (reported melting point = 135–137 °C [48]).

#### 4.1.2. General Procedure for the Synthesis of 6-amino-4-(aryl)-2-oxo-1-(3-(trifluoromethyl)phenyl)-1,2-dihydropyridine-3,5-dicarbonitriles (5a-f)

In ethanol (20 mL), equimolar quantities of intermediate 3 (2.28 g, 10 mmol), aromatic aldehydes **4a-f** (10 mmol), and malononitrile (0.66 g, 10 mmol) were mixed together, and then a few drops of piperidine were added to the mixture. The mixture was refluxed for a period of six hours. Following the completion of the reaction, the solid was allowed to cool and then filtered. The obtained material was crystallized with ethanol in order to produce 2-oxo-pyridine-3,5-dicarbonitriles **5a-f**, with yields ranging from 75% to 95%.

#### 4.1.3. General Procedure for the Synthesis of 4,7-dioxo-5-(aryl)-8-(3-(trifluoromethyl)phenyl)-3,4,7,8-tetrahydropyrido[2,3-d]pyrimidine-6-carbonitriles (6a-b)

Compounds **5b** or **5d** (10 mmol) were dissolved in 20 mL of formic acid, and concentrated sulfuric acid was added in a catalytic quantity for the purpose of promoting the reaction. The reaction mixture was refluxed for a period of between 4 and 8 h (under TLC monitoring). Following the completion of the reaction, the reaction mixture was allowed to cool to room temperature. The separated product was then filtered, washed with methanol, and crystallized from DMF in order to produce compounds **6a-b**.

#### 4.1.4. General Procedure for the Synthesis of 5-(4-(aryl)-2-methyl-4,7-dioxo-8-(3-(trifluoromethyl)phenyl)-3,4,7,8-tetrahydropyrido[2,3-d]pyrimidine-6-carbonitriles (7a-e)

A solution consisting of 6-amino-4-(aryl)-2-oxo-1-(3-(trifluoromethyl) phenyl)-1,2-dihydropyridine wer-3,5-dicarbonitriles (**3a**, **3b**, **3d-f**) with a concentration of 2 mmol in 10 mL of acetic anhydride were refluxed for 3–6 h. The solvent was then concentrated using decreased pressure, and the reaction mixture was poured into ice water (40 mL) to generate a solid precipitate. This precipitate was then filtered out, and target molecules **7a-e** were obtained by recrystallization from petroleum ether 60/80.

Spectral and elemental analyses for all the newly prepared molecules (**5a-f**, **6a-b**, **7**, **a-e**) are described in the Appendix A.

### 4.2. Cytotoxicity Assay

#### 4.2.1. Chemicals and Reagents

The following were purchased from Merck (Darmstadt, Hesse, Germany): paclitaxel (PTX/Taxol), 3-(4,5-dimethylthiazol-2-yl)-2,5 diphenyltetrazolium bromide (MTT), phosphate-buffered saline (PBS), culture medium (Roswell Park Memorial Institute; RPMI 1640, Buffalo, NY, USA), fetal bovine serum (FBS), trypsin/EDTA mixture, penicillin G, Str dye (St. Louis, MO, USA).

#### 4.2.2. Cell Culture

In vitro testing was performed on HepG2 and MCF-7 cell lines to determine the antitumor activity of the newly synthesized chemical compounds. Both cell lines originally came from Egypt’s National Cancer Institute (NCI), housed at Cairo University. At 37 °C with 5% carbon monoxide, the cells were grown in RPMI 1640 containing 10% fetal bovine serum, 50 micrograms per milliliter of streptomycin, and 100 international units per milliliter of penicillin. After trypsinization with 0.05% trypsin-EDTA, cell growth and morphology were evaluated, and passages were made when cells reached 80–90% confluence.

#### 4.2.3. Cell Viability Assay

The MTT assay protocol [49,50,51] is described in the Appendix A.

### 4.3. In Vitro VEGFR-2 and HER-2 Inhibition Assays

Assessment was performed using a VEGFR-2 (human) SimpleStep ELISA kit (ab213476) and HER-2 (human) SimpleStep ELISA kit (ab100510) (abcam^®^ Cambridge, CB2 0AX, Cambridge, UK) according to manufacturer instructions. In brief, tissue culture supernatant was added to the anti-tag antibody-coated wells, followed by the secondary antibody mix. To remove unbound material, the wells were washed after incubation. Adding tetramethyl benzidine (TMB) as a substrate and then allowing horseradish peroxidase (HRP) to catalyze the reaction resulted in a blue coloration during incubation. The final step in the transformation from blue to yellow was the addition of a stop solution, which halted the reaction. The intensity of the signal generated at 450 nm [52,53,54] was proportional to the amount of bound analyte.

### 4.4. Molecular Docking Studies

All the docking studies in the current work were carried out by implementing the docktite wizard of the Molecular Operating Environment (MOE 2019.02) package [55,56]. The structural coordinates of the target enzymes were downloaded from the protein data bank: PDB IDs 4ASD and 3RCD for VEGFR-2 in complex with sorafenib and HER-2 in complex with TAK-285, respectively. The two tested ligands were sketched using the MOE builder and then energy minimization and a conformational search were conducted using the default parameters of MOE software. The energy-minimized compounds **5a** and **5e** were saved to a single database file with the *mdb extension ready for docking conduction. By choosing the pocket around the binding domain of the co-crystalized ligands, the binding sites in the two targets were identified. Pose-retrieval docking studies for the X-ray coordinates of the co-crystalized ligands with their respective binding sites were conducted to verify the applied docking procedure [57]. After that, compounds **5a** and **5e** were docked into the two targets with the validated protocol. Finally, docking results were further analyzed through 2D interaction diagrams generated for the predicted binding mode between **5a** and **5e** with VEGFR-2 and HER-2 enzymes.

### 4.5. Molecular Dynamics

The four complexes **5e**–HER-2, **5e**–VEGFR-2, TAK285–HER-2, and sorafenib–VEGFR-2, in addition to the Apo form of HER-2 and VEGFR-2, were subjected to molecular dynamic simulation (MDS) for 150,000 ps. The same MDS methodology as previously published by our group was applied in the current study [58] (see Appendix A section for further details). Indicative parameters, such as RMSD and RMSF, were calculated to evaluate the stability of all the complexes.

#### MM-PBSA Calculation and Per-Residue Contribution

The binding free energies of the four complexes containing **5e**, TAK-285 and sorafenib were calculated using the MM-PBSA approach. The freely available g-mmpbsa package developed by Kumari et al. was used to process the calculations. In addition, the same package was utilized to decompose the binding free energies of **5e** with HER-2 and VEGFR-2 in terms of per-residue contributions (further details are provided in the Appendix A).

## 5. Conclusions

Thirteen target non-fused and fused pyridine-based derivatives **5a-f**, **6a-b**, and **7a-e** were designed and synthesized as novel antitumor molecules with potential VEGFR-2 and HER-2 inhibitory activities. The IC_50_ was evaluated with the MTT assay for all the synthesized compounds (**5a-f**, **6a-b**, and **7a-e**) against breast cancer (MCF-7) and hepatocellular carcinoma (HepG2) cell lines. The non-fused pyridone-based derivative **5** showed remarkable activity against both cell lines compared to the fused pyridopyrimidine derivatives **6** and **7**. In particular, non-fused pyridones **5a** and **5e** (IC_50_ = 1.77 ± 0.1 and 1.39 ± 0.08 µM, respectively) showed the best activity against MCF-7 cells. Furthermore, pyridones **5a** and **5e** elicited good inhibitory activities against both VEGFR-2 (IC_50_ = 0.217 ± 0.02 and 0.124 ± 0.011, respectively) and HER-2 kinases (IC_50_ = 0.168 ± 0.009 and 0.077 ± 0.003, respectively). The results of the docking analysis showed excellent docking scores and interesting binding modes and interactions for compounds **5a** and **5e** to VEGFR2 and HER-2 targets, which correlates with the in vitro kinase inhibitory activity results. Further endorsement for the proposed binding mode of the newly synthesized compound **5e** was conducted using MDs and the results revealed the formation of a highly stable complex for compound **5e** with HER-2 and VEGFR-2, as evidenced by the RMSD, RMSF, and free binding energy calculations. Notably, incorporation of the 2,4-dichlorophenyl moiety at the C-4 of the pyridine motif increased the binding affinity towards the active sites of the target enzymes and resulted in the best docking results and remarkable in vitro VEGFR-2 and HER-2 inhibition. Collectively, the scaffolds provided here could be a viable starting point for further investigation and development of effective anticancer candidates with dual VEGFR-2 and HER-2 inhibitory action.

## Data Availability

Data is contained within the article and Appendix A.

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
