# Peer review of "Identification of Novel Cyanopyridones and Pyrido[2,3-d]Pyrimidines as Anticancer Agents with Dual VEGFR-2/HER-2 Inhibitory Action: Synthesis, Biological Evaluation and Molecular Docking Studies"

_pharmaceuticals, 2022, doi:10.3390/ph15101262_

Round 1
Reviewer 1 Report
The authors have synthesized novel cyanopyridones and pyrido[2,3-d]pyrimidines and tested it as anticancer agents with dual VEGFR-2/HER-2 inhibitory action: According to presented results some of the compounds showed significant anticancer activity which allows further design of new generation of these types of compounds. The authors also commented influence of electron donating and withdrawing substituents on aryl ring towards anti cancer activity. Although all analytical data are clearly presented (IR, 1H NMR, 13C NMR, MS, etc.) there is a few points that needs to addressed before article can be suitable for publishing in Pharmaceuticals. For intermediate 3 references 47 and 48 are cited but at least melting points need to be stated and compared with literature data. In supporting information document all 13C NMR needs to be carefully checked. For example, for compound 5a 18 signals are reported but according to chemical structure only 17 or 19 are possible. According to 1H NMR data symmetry of Ph ring is seen due to free rotation and as a result 13C NMR could have only 17 signals. All other compounds needs to be checked similar way.
Author Response
we truly thank the reviewer for his/her professional handling of our manuscript.
please see the attachment for responses

Reviewer 2 Report
The paper shows the results about the synthesis, biological evaluation and molecular docking of new pyridones and a Pyridopyrimidines fused system.
The paper is well structured. The introduction is complete and the results are very clearly presented. The compounds showed a promising anti-cancer activity and represent a new scaffold to be manipulated in order to obtain new drugs.
Line 1: When fused systems are named, the base component locant should be lowercase.
Author Response
we truly thank the reviewer for his/her professional revision of our manuscript
comment
Line 1: When fused systems are named, the base component locant should be lowercase.
response:
the base component locant was corrected as the reviewer requested
Reviewer 3 Report
Based on the chemical structure of known anticancer molecules, author desinged and synthesized series of pyridone derivatives using simple "one -pot" heterocycle fusing reactions. They hope to find novel anticancer drug candidates possessing dual inhibitory action against VEGFR-2 and HER-2, because VEGFR-2 and HER-2 play cruical roles in the growth and metastasis of cancer cells. Authors test the cytotoxicity of all synthesized compounds using MTT assay and found 5a and 5e are better than controls, also they analysized SAR. They further investigated the inhibition ability of 5a, 5e against VEGFR-2 and HER-2 kinases and they found 5e are better than control. Authors rationalized the activity of 5a, 5e using molecular docking. Authors finally carried out the molecular dynamics, RMSD and RMSF analysis and free energy calculation for 5e.
Minor improvement need to addressed to make it publishable
1) Please list yield for each compound in the inset table of scheme 1 and 2; 2) please enlarge the inside labels of Fig 11, 12 and 13 to make them visible; 3) Please integrate each group peaks of 1H NMRs if the splitting is well. DON'T INTEGRATE EVERYTHING TOGETHER! 4) Please list the J value (1H NMR) for each group peaks splitting well (doublet, etc.)
Author Response
We truly thank the reviewer for his/her professional handling of our manuscript.
Please list yield for each compound in the inset table of scheme 1 and 2; 2)
reponse: we thank the reviewer for such comment, we have listed the yield for each compound in the inset table of scheme 1 and 2.
2) Please enlarge the inside labels of Fig 11, 12 and 13 to make them visible
response: thanks a lot for such notice, we have enlarged all the figures' labels as requested
3) Please integrate each group peaks of 1H NMRs if the splitting is well. DON'T INTEGRATE EVERYTHING TOGETHER!
response: we appreciate the reviewer's critique and the peaks of 1H NMRs were properly integrated
4) Please list the J value (1H NMR) for each group peaks splitting well (doublet, etc.)
response: thanks a lot for such a comment, the j value for each well-split peak was provided
Reviewer 4 Report
Al-Warhi et al. synthesized a series of cyanopyridone derivatives that demonstrated antitumor inhibitory activities in breast and liver cancer cells. The authors then performed molecular docking and simulations study on selected promising compounds to elucidate their binding modes and rationalized the observed biological activities. These inhibitors could be potentially interesting, and their inhibitory potency can be further developed. However, the manuscript needs to be re-worked extensively, in particular the overall language needs to be revised and some claims need to be substantiated or clarified. There are also several issues that the authors should address before the paper can be considered for publication.
Line 40: investigations to elucidate the molecular binding mode
Line 53: Should be “kinases”
Line 53-55: Rephrase, sentence is unclear.
Line 61: …an even better antitumor activity was observed after combining both HER-2 and VEGFR-2 inhibitors
Line 69: include the word “the”
Line 71: neratinib word should be non-capitalized
Lines 76-79: Please provide reference.
Line 83: Should be “analogs”
Line 100: Same? Or similar?
Line 108: Should be “newly”
Line 112: Please could the authors clarify what is meant by “firm active sites”?
Line 117: “such a group”… “form a hydrogen bonding interaction”
Line 163-164: to measure the potential cytotoxicity of?
Lines 165, 173: Missing the word “values”. IC50 values
Line 188: “did not”, not “haven’t”
Line 195, 196: remove the word “was”
Line 198: showed greater anticancer activity
Line 221: Please indicate what the values represent? Mean ± S.D. of how many experiments?
Lines 228, 245, 256, 276: Should be S.D.
Line 229: taxol word should be non-capitalized
Figure 5: Should be 5a, 5e on the x-axis, not 3a, 3e
Line 313: What do the authors mean by a perfect fit into the active site? Please substantiate the claim, or rephrase if possible.
Line 314: Please include more details about the binding interactions. What is the nature of the interaction with Cys1045?
Lines 328-329: Edit sentence for clarity.
Lines 330 and 334: trifluoromethyl moiety/group. The word moiety (or group) is missing.
Line 332: What is the nature of the interaction with Leu796?
Line 335 and 336: Typological errors in “Kacl/mole” and “Kcal”. Please correct them.
Line 339: What do the authors mean by a perfect fit into the binding site?
Lines 340-341: Provide more information on nature of interactions.
Line 349: Include more details about the binding interactions. What is the nature of the interaction with Asp863 and Gly865?
Line 454: inhibitory activities
Line 458: Edit the sentence. “Interprets” is not an appropriate word to use in this case.
Line 460: revealed the formation of a highly stable kinase-compound complex for…
In Figure 4, VEGFR-2 and HER-2 should be used for consistency throughout the manuscript.
Captions for Figures 7/8/9 could be more informative. For example, what do the differently colored arrows mean? Why are the amino acid residues colored differently? What do blue halos/shading represent? Describing the figures in more details will help the readers understand the figures more clearly. Alternatively, presenting the data in in the form of a 3D protein-compound complex structure would be visually more informative.
Why was taxol used in the MTT assay, but lapatinib was used in the kinase assay as reference? Could lapatinib be used in the MTT assay?
Please be consistent in reporting the IC50 values throughout the manuscript (tables, text). Keep to 3 significant figures, or 2 decimal places when possible.
Please be consistent in using either the word “synthesized” or “synthesised”.
Please consistently ensure compound numbers are in bold throughout the manuscript.
Please ensure words are capitalized when appropriate (i.e. if it is the first word in a sentence).
Please ensure PDB IDs are all capitalized for consistency (lines 327, 356, 547).
Please ensure all the references are of consistent format.
In Supplementary Information (SI), under <Cell viability assay> section (page 7 in SI), please correct the various errors and clarify some points.
- Cell density. Should be 1 × 104 cells/well.
- 100 L of MTT? Or 100 μL of MTT?
- After adding 100 L of acidified isopropanol to each well? Or 100 μL?
- Inhibitory concentrations (IC50): Should be “Half-maximal inhibitory concentrations (IC50)….”
- What software was used to fit the dose response curves and what model (3 parameter or 4 parameter logistic fit or others?) was used to derive the IC50 values?
Under <Molecular dynamics> section (page 7 in SI), please correct the error.
- Compound 3e? Or should it be compound 5e?
Author Response
we truly appreciate the reviewer for his/her professional handling of our manuscript. indeed, the raised points by the reviewer significantly enhanced the quality of our manuscript.
Please see the attachment

Round 2
Reviewer 4 Report
Changes accepted